# U.S. household food acquisition behaviors during the COVID-19 pandemic

**Brenna Ellison** [1]*, **Melissa Ocepek**[2], **Maria Kalaitzandonakes**[3]

**1** Department of Agricultural Economics, Purdue University, West Lafayette, Indiana, United States of America, **2** School of Information Sciences, University of Illinois at Urbana-Champaign, Urbana, Illinois, United States of America, **3** Department of Agricultural and Consumer Economics, University of Illinois at Urbana-Champaign, Urbana, Illinois, United States of America

* bdelliso@purdue.edu

**Data Availability Statement:** All relevant data are within the article and its Supporting information files.

**Funding:** This research was supported by funding from the National Institute of Food and Agriculture,

## Abstract

The COVID-19 pandemic upended how many Americans acquire foods. In this paper, we analyze eight food acquisition activities at different points in the pandemic, which allows us to evaluate how food acquisition changed as case rates changed and vaccine rollouts occurred. We collected data from three nationally representative online samples in September 2020, December 2020, and March 2021. We evaluate changes across time and across demographics using a multivariate probit model. Across time, we find that in-person grocery shopping remained extremely common (over 90%) throughout the pandemic. Food acquisition activities with less in-person contact (e.g., ordering from a meal kit service, online grocery shopping) peaked in December 2020, likely due to the surge in cases during that period. Ordering take-out from a restaurant remained common throughout the pandemic, but indoor dining increased significantly in March 2021 when vaccines were becoming more widely available. Food acquisition activities also varied across consumer groups, particularly indoor and outdoor restaurant dining. Overall our results offer evidence that in-person grocery shopping is a staple food acquisition activity that is unlikely to be changed; however, there is a segment of consumers who complement their in-person grocery shopping with online grocery shopping options. Further, relative to grocery stores, restaurants may be more vulnerable to surges in COVID-19 case rates. We conclude with implications for grocery retailers and restaurants as they continue to navigate operational challenges associated with the COVID-19 pandemic.

## Introduction

In March 2020 the world changed as the COVID-19 pandemic spread quickly across the globe. Throughout that month millions of Americans' lives were dramatically altered as they were asked and ordered to stay at home. Workplaces, shopping centers, and schools shut down, restaurants and grocery stores moved to or added delivery and curbside pick-up. Toilet paper and other product shortages resulted in panic buying and stockpiling that made pandemic-induced supply chain problems worse [1, 2]. Short term changes quickly became the new normal as many Americans continued to be unsure how to protect themselves from a virus that

under award number ILLU-470-334 at the University of Illinois at Urbana-Champaign (author: BE). The funders had no role in study design, data collection and analysis, decision to publish, or preparation of the manuscript.

**Competing interests:** The authors have declared that no competing interests exist.

they did not understand. Grocery shopping and procuring food from a restaurant became more stressful than ever before, with the FDA, CDC, and major news outlets all offering recommendations and tips on how to acquire food safely [3–5].

The COVID-19 pandemic upended food acquisition activities for many U.S. consumers, particularly at the early stages of the pandemic. However, less is known about how food acquisition behaviors changed over the course of the pandemic as COVID-19 case rates fluctuated and vaccines became available. In this paper, we investigate prevalence rates for eight food acquisition activities (in person grocery shopping; online grocery shopping; ordering from a meal kit service; indoor restaurant dining; outdoor restaurant dining; ordering take-out from a restaurant; visiting a food bank; and visiting a farmer's market) among U.S. households at three different points in the pandemic: September 2020, December 2020, and March 2021. In addition, we explore potential heterogeneity in food acquisition activities across household characteristics.

Prior to the pandemic, Americans spent over half of their food expenditures on food away from home (FAFH) [6]. In 2019, full service and limited-service restaurants accounted for $717.4 billion of food expenditures [7]. However, the rapid spread of COVID-19 caused many states to restrict access to restaurants—in particular, indoor dining—early in the pandemic. Zeballos & Dong [8] found significant reductions in FAFH spending in 2020 among U.S. households, and Yenerall et al. [9] reported much lower rates of indoor and outdoor restaurant dining in June 2020 relative to ordering restaurant take-out. The State of the Restaurant Industry report [10] highlighted the profound shifts restaurateurs made in an attempt to increase take-out and outdoor dining capacity. They found that 80 percent of full-service restaurants (fine dining, family dining and fast casual) added a curbside pickup option and slightly less than half added delivery. They also found that 62% of fine dining, 56% of casual dining, and 48% of family dining restaurants expanded their outdoor capacity in 2020.

Grocery shopping was also nearly ubiquitous before the pandemic. In 2019, grocery stores, warehouse clubs, and supercenters accounted for $648.4 billion of food expenditures [7]. Food at home (FAH) expenditures significantly increased during the first year of the COVID-19 pandemic [8]. While the large majority of U.S. consumers shop for groceries in person, the 2019 U.S. Grocery Shopper Trends industry report noted that online grocery shopping was increasing [11]. Early research suggests that the pandemic accelerated rates of online shopping in the U.S. and abroad [1, 12–16]. Rates of online shopping may have increased among food insecure households, in particular, as the USDA expanded access to online shopping options for Supplemental Nutrition Assistance Program (SNAP) recipients early in the pandemic [17]. Still, there is less evidence on whether the move to online shopping was a long-term change or a short-term adaptation for shoppers.

The use of meal kit services and food banks were also impacted by the COVID-19 pandemic. Meal kit services, much like online grocery shopping, allow for greater social distancing and experienced a significant increase in usage in the first year of the pandemic. Before the pandemic, less than 15% of households had ordered a meal kit; in 2020, almost 25% of households had tried a meal kit service [18]. Demand for food banks also increased significantly, driven by sharp increases in unemployment [19]. There is less evidence on how the pandemic impacted farmer's markets, but they likely also saw a boost as consumers were looking for safe, socially-distanced options.

This research makes two key contributions to the growing literature on household food behaviors during the COVID-19 pandemic. First, our study considers food acquisition behaviors at multiple points in the pandemic; earlier studies tend to present results from one cross-sectional survey, often from the first few months of the pandemic in 2020. As the pandemic has extended far longer than many expected, our research is well positioned to comment on

how households were acquiring foods at later points in the pandemic, particularly during the first big surge in Winter 2020 and the beginning stages of vaccine rollouts. Second, our study design allows us to examine heterogeneity in food acquisition behaviors across household characteristics. Thus, we are not only able to capture how aggregate acquisition behaviors change over time, but we can also see which subpopulations may be driving such changes.

## Methods

### Sample recruitment

To examine how U.S. household food acquisition behaviors changed over the course of the COVID-19 pandemic, we recruited three cross-sectional samples of U.S. consumers at three different points in time: September 2020; December 2020; and March 2021. These time points allowed us to capture household food acquisition behavior *after* the initial panic buying and stockpiling phases reported by early COVID-19 research (e.g., [2, 13–15, 20]). Additionally, these time intervals correspond with very different points in the trajectory of the pandemic. In September 2020, new COVID-19 case counts were relatively low in the U.S. (an average of approximately 40,000 new cases/day [21]) with concerns that cases would rise significantly during the winter months and uncertainty around the timing of vaccine availability. In December 2020, new cases averaged over 210,000/day [21] in what is now considered to be the first large surge in COVID-19 cases. The first COVID-19 vaccines became available to certain segments of the population in the U.S. in December [22]. By March 2021, new cases were falling (average of approximately 57,000 new cases/day [21]) with the expectation that widespread vaccine deployment would continue the downward trend (though with stagnating vaccination rates and the rise of the Delta and Omicron variants, we now know that not to be the case).

In each cross-sectional wave of the survey, we recruited approximately 1,000 U.S. consumers to complete an online survey in the Qualtrics survey platform. We did not recruit the same individuals to participate in each wave of the survey; each sample was recruited to be representative of the U.S. population in terms of gender, age, income, and geographic region. Individuals were eligible to participate if they were 18 years of age or older and responsible for at least 50% of the grocery shopping in their household. Respondents provided written consent by answering affirmatively that they would like to participate in the study in the first question of the online survey. Survey completion time was less than 15 minutes. We worked with Qualtrics to manage participant recruitment through an existing consumer panel and payment distribution. The study was approved by the University of Illinois at Urbana-Champaign Institutional Review Board (IRB #21186).

### Survey design

The key question of interest from this survey related to how households acquired food throughout the COVID-19 pandemic. More specifically, we asked participants: "In the last 14 days (two weeks), have you completed any of the following activities?" The survey included a total of eight food acquisition activities: 1) shopping for groceries in person; 2) shopping for groceries online; 3) ordering from a meal kit service (e.g., Blue Apron, Hello Fresh); 4) eating at a restaurant and sitting indoors; 5) eating at a restaurant and sitting outdoors; 6) ordering take-out from a restaurant; 7) visiting a food bank; and 8) visiting a farmer's market. For each food acquisition activity, participants could answer 'Yes', 'No', or 'I don't remember'. Responses of 'No' and 'I don't remember' were combined for analysis purposes.

Participants were also asked to complete a series of sociodemographic questions, including questions related to gender, age, income, education, race, geographic region (based on four U. S. census regions: Northeast, South, Midwest, and West), metropolitan status (e.g., whether or

not they lived in a densely-populated metropolitan area), political affiliation, and nutrition assistance recipient status (whether they were receiving any benefits from nutrition assistance programs like SNAP, WIC, the National School Lunch Program, etc. at the time of survey completion).

### Data analysis

To get a general sense of trends and changes in food acquisition behaviors over the course of the COVID-19 pandemic, we first calculated the raw proportions of participants that engaged in each food acquisition activity in each of the three survey waves. For analysis purposes, we excluded participants who skipped the entire set of food acquisition activities as well as participants who straightlined answers (responded with all 'Yes' or all 'No') across the eight food acquisition activities as it is unlikely that a given individual completed all or none of these activities in the prior 14-day period.

While general trends are informative, there was also likely heterogeneity in acquisition activities—for example, one's age (as a proxy for likelihood of severe infection) or geographic region may have been associated with engaging in or avoiding certain acquisition activities more than others. To that end, we explore potential heterogeneities in food acquisition behaviors with the following model:

$$FAA_{ij} = f\left(Dec_j, \ Mar_j, \ X_i, Dec_j * X_i, Mar_j * X_i\right) \tag{1}$$

where $FAA_{ij}$ is coded as one if individual $i$ chose to engage in food acquisition activity $j$ and zero otherwise. $Dec_j$ and $Mar_j$ are indicator variables for the December 2020 and March 2021 survey waves, respectively; effects should be interpreted relative to the September 2020 survey wave. $X_i$ is a vector or demographic variables including gender, age, income, education, race, geographic region, metropolitan status, political affiliation, and nutrition assistance recipient status. $Dec_j * X_i$ and $Mar_j * X_i$ represent interaction effects between the December 2020 and March 2021 survey waves and sociodemographic variables.

Because engaging in certain food acquisition activities may be more or less related to engaging in others (for example, individuals who choose to shop for groceries online may be more likely to order from online meal kit services), we estimated Eq (1) using a multivariate probit model. This type of model allows for simultaneous estimation of Eq (1) for all eight food acquisition activities and estimates pairwise correlations across the errors of the eight equations [23]. The null hypothesis in the multivariate probit model is that the error terms across equations are uncorrelated; in this case, individual probit models can be estimated for each food acquisition activity. However, if correlations are significant, the multivariate probit model is preferred. We performed a Likelihood Ratio to determine the appropriate model.

### Results

Table 1 provides sample characteristics for each of the three survey waves. In total, 2,793 responses were retained for analysis. While each sample was individually recruited to match the U.S. population on gender, age, income, and geographic region, there were some differences across waves due to the exclusion of straightlining responses. Specifically, the December 2020 wave of the survey had a significantly higher rate of straightlining behavior, resulting in a final sample of 886 participants.

Fig 1 presents the proportion of respondents who reported engaging in each food acquisition activity by survey wave. Shopping for groceries in person was the most performed food acquisition activity, with 90% or more of participants engaging in this activity in each of the

**Table 1. Summary of sample characteristics by survey wave.**

| Variable | Sample Proportion (%) | | |
|---|---|---|---|
| | September 2020 (n = 965) | December 2020 (n = 886) | March 2021 (n = 942) |
| Gender* | | | |
| Male | 41.8% | 44.2% | 46.8% |
| Female | 58.1% | 55.1% | 52.4% |
| Non-binary/third gender | 0.1% | 0.7% | 0.7% |
| Age | | | |
| 18–34 years | 30.0% | 30.0% | 29.8% |
| 35–54 years | 33.9% | 32.2% | 33.6% |
| 55 years or older | 36.2% | 37.8% | 36.6% |
| Household income | | | |
| Less than $50,000 | 40.1% | 43.4% | 41.2% |
| $50,000—$99,999 | 32.8% | 33.9% | 34.3% |
| $100,000 or more | 27.2% | 22.7% | 24.5% |
| Education: Bachelor's degree or higher* | 57.4% | 46.4% | 47.9% |
| Race* | | | |
| White | 83.2% | 81.6% | 83.7% |
| Black or African American | 6.7% | 10.3% | 6.5% |
| Asian | 6.9% | 3.1% | 4.8% |
| Other | 3.1% | 5.1% | 5.0% |
| Geographic region | | | |
| Northeast | 21.5% | 16.5% | 17.4% |
| South | 38.0% | 38.3% | 37.8% |
| Midwest | 19.6% | 22.4% | 22.7% |
| West | 20.9% | 22.9% | 22.1% |
| Lives in metropolitan area | 40.7% | 44.9% | 39.7% |
| Political affiliation* | | | |
| Republican | 32.2% | 29.7% | 29.2% |
| Democrat | 36.6% | 43.9% | 38.8% |
| Independent/Other | 31.2% | 26.4% | 32.06% |
| Nutrition assistance recipient (SNAP, WIC, etc.)* | 13.7% | 29.8% | 21.9% |

Notes: Some percentages may not add to 100% due to rounding. An * denotes a significant difference in the distribution of a sociodemographic variable across the three survey waves (Chi-squared p-value < 0.05).

three survey waves. Ordering take-out from a restaurant was another common food acquisition activity for participants across the three waves, performed by 62.9%, 70.0%, and 68.9% of participants in September 2020, December 2020, and March 2021, respectively. Other food acquisition activities were reported at lower rates; visiting a food bank, visiting a farmer's market, and ordering from a meal kit service exhibited the lowest shares of engagement.

Looking at overall trends in food acquisition activities across time, Fig 1 shows that several food acquisition activities experienced increases in December 2020 relative to the September 2020 and March 2021 survey waves. We observed this trend for shopping for groceries online, visiting a food bank, ordering from a meal kit service, and visiting a farmer's market. Ordering take-out from restaurants also peaked in December 2020, yet this was a more common activity across all waves. Restaurant dining exhibited different trends compared to other food acquisition activities. Rates of indoor and outdoor restaurant dining were nearly identical in September 2020 (31.8% for indoor dining; 31.3% for outdoor dining), but their paths diverged as the

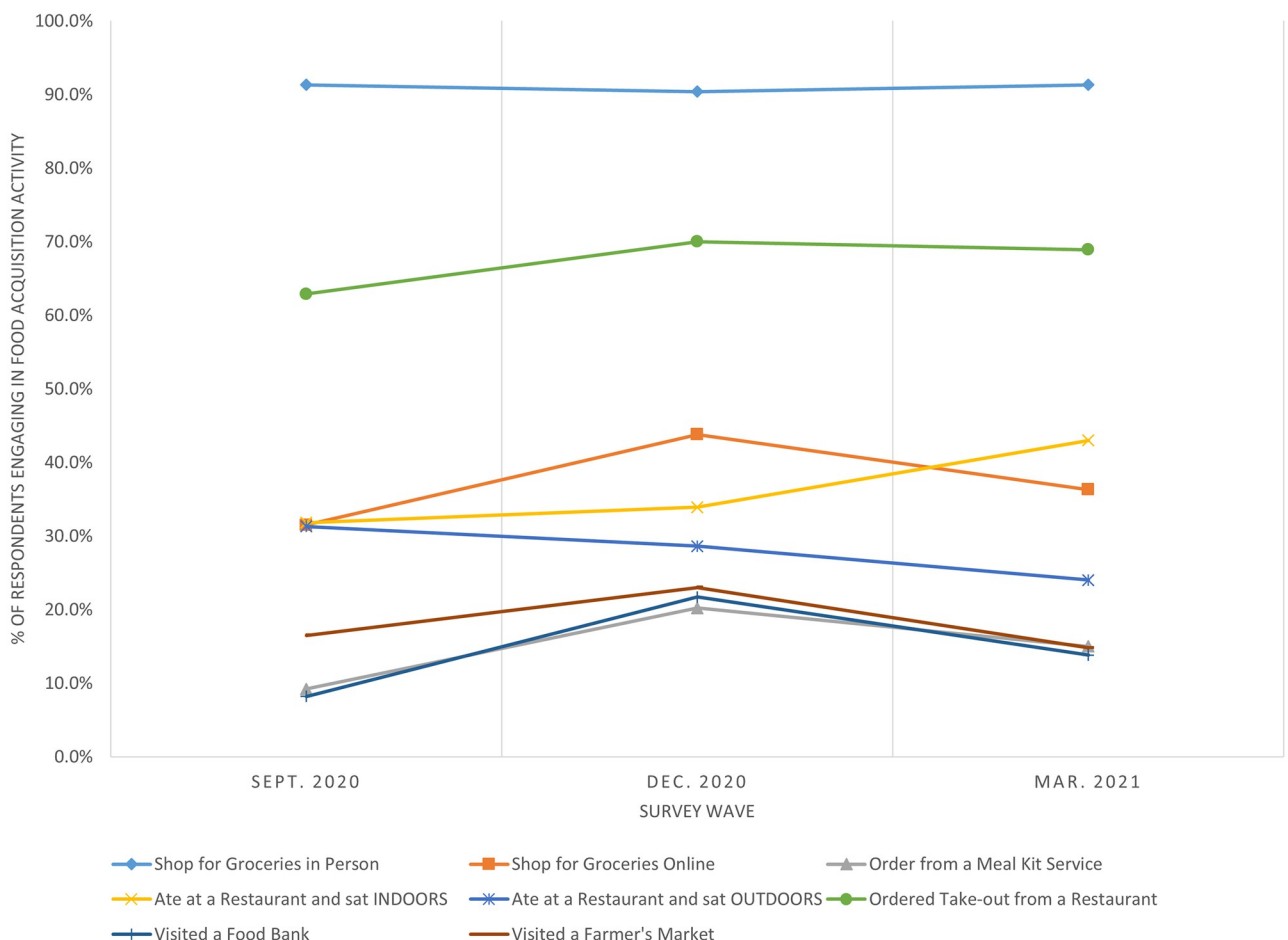

**Fig 1. Reported shares of food acquisition activities performed by survey wave.**

U.S. moved through the pandemic. The proportion of respondents engaging in indoor restaurant dining increased across the three survey waves, with a notable increase in the March 2021 survey wave. By contrast, outdoor restaurant dining experienced a more consistent decline across the three survey periods.

While Fig 1 provides some perspective on how food acquisition activities changed over time, it does not consider how different subpopulations may have adjusted their food acquisition activities over the course of the pandemic. Table 2 presents the results from the multivariate probit model that explores these potential heterogeneities. A Likelihood Ratio test revealed significant correlations across food acquisition activities; thus, the multivariate probit was the preferred model.

Table 2 reveals that the probabilities of shopping online for groceries, ordering from a meal kit service, and visiting a food bank significantly increased in December 2020 relative to September 2020; further, significant increases were sustained into March 2021 for online grocery shopping and visiting a food bank, all else constant. The probability of dining at a restaurant and sitting indoors was significantly higher in March 2021 relative to September 2020, confirming the visual trend observed in Fig 1.

In terms of heterogeneity in acquisition activities, we observed significant differences based on the age of participants. In particular, the youngest individuals (ages 18–34) were

**Table 2. Multivariate probit estimation results for eight food acquisition activities.**

| | Food Acquisition Activity (= 1 if activity performed in last 14 days; = 0 otherwise) | | | | | | | |
|---|---|---|---|---|---|---|---|---|
| | **Grocery Shop in Person** | **Grocery Shop Online** | **Order Meal Kit** | **Indoor Restaurant Dining** | **Outdoor Restaurant Dining** | **Restaurant Take-out** | **Visit Food Bank** | **Visit Farmer's Market** |
| December2020 (Dec)[a] | -0.493 | 0.663** | 0.775* | 0.226 | 0.034 | -0.221 | 1.088** | 0.051 |
| | (0.468) | (0.311) | (0.442) | (0.326) | (0.315) | (0.310) | (0.432) | (0.351) |
| March 2021 (Mar)[a] | -0.259 | 0.563* | 0.357 | 0.505* | -0.079 | -0.313 | 1.111** | -0.133 |
| | (0.448) | (0.297) | (0.436) | (0.301) | (0.304) | (0.293) | (0.429) | (0.353) |
| Female[b] | 0.084 | -0.119 | -0.145 | 0.049 | -0.124 | -0.085 | -0.252* | -0.071 |
| | (0.123) | (0.090) | (0.124) | (0.089) | (0.088) | (0.086) | (0.136) | (0.101) |
| Dec*Female | -0.129 | 0.010 | -0.468** | -0.161 | -0.026 | 0.094 | -0.208 | -0.087 |
| | (0.183) | (0.133) | (0.181) | (0.136) | (0.139) | (0.133) | (0.181) | (0.152) |
| Mar*Female | -0.188 | -0.037 | -0.347* | -0.168 | -0.066 | 0.069 | -0.170 | -0.347** |
| | (0.183) | (0.133) | (0.188) | (0.128) | (0.138) | (0.129) | (0.189) | (0.159) |
| Inc50-99K[c] | -0.333** | 0.160 | 0.256* | 0.027 | 0.130 | 0.199* | 0.025 | 0.065 |
| | (0.145) | (0.106) | (0.152) | (0.106) | (0.106) | (0.103) | (0.162) | (0.120) |
| Dec*Inc50-99K | 0.511** | 0.211 | 0.043 | 0.123 | -0.048 | -0.086 | -0.343 | 0.255 |
| | (0.209) | (0.152) | (0.222) | (0.156) | (0.162) | (0.152) | (0.215) | (0.178) |
| Mar*Inc50-99K | 0.199 | 0.065 | -0.049 | 0.131 | 0.033 | -0.212 | -0.188 | 0.321* |
| | (0.207) | (0.150) | (0.215) | (0.147) | (0.156) | (0.149) | (0.219) | (0.180) |
| Inc100K+[c] | -0.141 | 0.213* | 0.456** | 0.127 | 0.193 | 0.169 | 0.112 | 0.127 |
| | (0.176) | (0.126) | (0.178) | (0.121) | (0.119) | (0.117) | (0.205) | (0.137) |
| Dec*Inc100K+ | 0.449* | 0.345* | 0.145 | 0.533*** | 0.140 | -0.176 | 0.115 | 0.452** |
| | (0.270) | (0.193) | (0.261) | (0.194) | (0.196) | (0.194) | (0.269) | (0.214) |
| Mar*Inc100K+ | 0.028 | 0.256 | 0.078 | 0.222 | 0.142 | -0.163 | -0.020 | 0.298 |
| | (0.257) | (0.182) | (0.252) | (0.174) | (0.183) | (0.175) | (0.271) | (0.217) |
| Age18-34[d] | -0.253 | 0.912*** | 0.850*** | 0.186 | 0.243* | 0.244** | 0.621*** | 0.504*** |
| | (0.169) | (0.128) | (0.180) | (0.125) | (0.126) | (0.122) | (0.209) | (0.144) |
| Dec*Age18-34 | 0.082 | -0.438** | 0.497* | 0.409** | 0.606*** | 0.414** | -0.411 | 0.297 |
| | (0.227) | (0.172) | (0.262) | (0.174) | (0.178) | (0.172) | (0.251) | (0.198) |
| Mar*Age18-34 | -0.116 | -0.152 | 0.562** | 0.359** | 0.599*** | 0.546*** | -0.304 | 0.143 |
| | (0.237) | (0.176) | (0.267) | (0.170) | (0.182) | (0.174) | (0.265) | (0.208) |
| Age35-54[d] | 0.007 | 0.526*** | 0.307* | -0.118 | 0.004 | -0.012 | 0.274 | 0.309** |
| | (0.160) | (0.115) | (0.170) | (0.111) | (0.110) | (0.106) | (0.206) | (0.128) |
| Dec*Age35-54 | 0.057 | -0.191 | 0.753*** | 0.404** | 0.420** | 0.536*** | -0.128 | 0.294 |
| | (0.233) | (0.166) | (0.259) | (0.168) | (0.174) | (0.163) | (0.252) | (0.193) |
| Mar*Age35-54 | -0.162 | -0.099 | 0.494** | 0.286* | 0.382** | 0.323** | -0.036 | -0.120 |
| | (0.230) | (0.161) | (0.251) | (0.153) | (0.166) | (0.151) | (0.255) | (0.194) |
| Northeast[e] | 0.243 | -0.097 | 0.006 | 0.358*** | 0.347*** | -0.071 | 0.080 | 0.405*** |
| | (0.210) | (0.138) | (0.186) | (0.138) | (0.129) | (0.129) | (0.235) | (0.151) |
| Dec*Northeast | -0.258 | -0.038 | 0.004 | -0.176 | -0.497** | 0.175 | 0.073 | -0.281 |
| | (0.272) | (0.198) | (0.253) | (0.203) | (0.199) | (0.197) | (0.288) | (0.216) |
| Mar*Northeast | -0.615** | 0.356* | 0.428* | -0.196 | -0.587*** | 0.194 | 0.047 | -0.092 |
| | (0.287) | (0.196) | (0.256) | (0.194) | (0.197) | (0.193) | (0.291) | (0.221) |
| South[e] | -0.230 | 0.148 | 0.007 | 0.372*** | -0.231* | 0.111 | 0.367* | 0.006 |
| | (0.170) | (0.123) | (0.172) | (0.126) | (0.123) | (0.119) | (0.200) | (0.144) |
| Dec*South | 0.627*** | -0.168 | -0.285 | -0.005 | 0.216 | -0.091 | -0.194 | -0.058 |
| | (0.232) | (0.171) | (0.230) | (0.177) | (0.176) | (0.171) | (0.246) | (0.197) |

(*Continued*)

**Table 2.** (*Continued*)

| | Food Acquisition Activity (= 1 if activity performed in last 14 days; = 0 otherwise) | | | | | | | |
|---|---|---|---|---|---|---|---|---|
| | Grocery Shop in Person | Grocery Shop Online | Order Meal Kit | Indoor Restaurant Dining | Outdoor Restaurant Dining | Restaurant Take-out | Visit Food Bank | Visit Farmer's Market |
| Mar*South | 0.165 | -0.127 | -0.025 | -0.230 | 0.106 | 0.016 | -0.628** | 0.111 |
| | (0.245) | (0.170) | (0.233) | (0.170) | (0.173) | (0.168) | (0.253) | (0.202) |
| Midwest[e] | -0.119 | 0.031 | -0.163 | 0.460*** | 0.003 | -0.159 | 0.223 | 0.161 |
| | (0.193) | (0.142) | (0.207) | (0.141) | (0.137) | (0.133) | (0.236) | (0.160) |
| Dec*Midwest | 0.624** | -0.273 | -0.270 | -0.551*** | -0.359* | 0.177 | -0.249 | -0.357 |
| | (0.269) | (0.193) | (0.280) | (0.201) | (0.201) | (0.189) | (0.287) | (0.224) |
| Mar*Midwest | -0.139 | -0.249 | -0.177 | -0.391** | -0.416** | 0.164 | -0.490* | -0.536** |
| | (0.269) | (0.196) | (0.283) | (0.190) | (0.198) | (0.187) | (0.293) | (0.241) |
| Republican[f] | 0.210 | -0.021 | 0.339** | 0.384*** | 0.452*** | -0.036 | 0.252 | 0.134 |
| | (0.149) | (0.112) | (0.162) | (0.108) | (0.111) | (0.106) | (0.172) | (0.124) |
| Dec*Republican | -0.130 | -0.006 | -0.192 | 0.031 | -0.416** | 0.117 | -0.014 | 0.022 |
| | (0.226) | (0.164) | (0.236) | (0.164) | (0.171) | (0.163) | (0.228) | (0.186) |
| Mar*Republican | 0.109 | -0.011 | -0.358 | -0.081 | -0.533*** | 0.177 | -0.165 | 0.164 |
| | (0.218) | (0.160) | (0.232) | (0.152) | (0.166) | (0.155) | (0.232) | (0.192) |
| Democrat[f] | 0.174 | 0.119 | 0.320** | -0.099 | 0.146 | 0.032 | 0.140 | 0.011 |
| | (0.144) | (0.105) | (0.155) | (0.108) | (0.108) | (0.103) | (0.167) | (0.122) |
| Dec*Democrat | -0.150 | 0.027 | -0.206 | 0.242 | -0.142 | -0.105 | 0.242 | 0.034 |
| | (0.206) | (0.153) | (0.217) | (0.160) | (0.162) | (0.153) | (0.216) | (0.178) |
| Mar*Democrat | 0.136 | 0.076 | -0.028 | 0.038 | -0.073 | -0.058 | 0.108 | 0.341* |
| | (0.206) | (0.151) | (0.213) | (0.151) | (0.158) | (0.150) | (0.222) | (0.184) |
| Bachelor's[g] | 0.201 | -0.035 | 0.185 | 0.029 | 0.260*** | 0.082 | -0.005 | 0.209* |
| | (0.134) | (0.098) | (0.140) | (0.097) | (0.098) | (0.095) | (0.153) | (0.111) |
| Dec*Bachelor's | -0.560*** | 0.255* | 0.124 | -0.326** | 0.039 | -0.188 | 0.040 | -0.082 |
| | (0.194) | (0.143) | (0.205) | (0.149) | (0.151) | (0.144) | (0.203) | (0.167) |
| Mar*Bachelor's | -0.287 | 0.299** | 0.371* | 0.037 | -0.106 | 0.053 | -0.072 | -0.179 |
| | (0.193) | (0.139) | (0.199) | (0.136) | (0.144) | (0.138) | (0.206) | (0.167) |
| Metro[h] | -0.027 | 0.154 | 0.397*** | -0.067 | 0.113 | -0.013 | 0.050 | -0.098 |
| | (0.129) | (0.094) | (0.130) | (0.093) | (0.093) | (0.090) | (0.147) | (0.107) |
| Dec*Metro | -0.017 | 0.144 | 0.010 | -0.124 | 0.059 | -0.038 | 0.023 | 0.203 |
| | (0.185) | (0.135) | (0.184) | (0.140) | (0.141) | (0.136) | (0.189) | (0.155) |
| Mar*Metro | -0.041 | -0.134 | -0.211 | -0.002 | 0.033 | 0.066 | 0.034 | 0.247 |
| | (0.186) | (0.135) | (0.183) | (0.131) | (0.138) | (0.133) | (0.194) | (0.159) |
| NutritionAssist[i] | -0.050 | 0.368*** | 0.552*** | 0.016 | -0.091 | -0.116 | 1.159*** | 0.203 |
| | (0.175) | (0.127) | (0.163) | (0.135) | (0.139) | (0.130) | (0.153) | (0.144) |
| Dec*NutritionAssist | -0.078 | -0.152 | -0.273 | 0.050 | 0.128 | -0.232 | -0.327* | 0.035 |
| | (0.222) | (0.165) | (0.207) | (0.172) | (0.177) | (0.169) | (0.190) | (0.183) |
| Mar*NutritionAssist | -0.255 | -0.042 | -0.225 | -0.060 | 0.207 | -0.220 | -0.076 | 0.128 |
| | (0.227) | (0.168) | (0.214) | (0.172) | (0.181) | (0.170) | (0.197) | (0.193) |
| White[j] | -0.620** | 0.360** | 0.323 | 0.337** | 0.049 | -0.325** | -0.066 | 0.062 |
| | (0.280) | (0.152) | (0.215) | (0.165) | (0.150) | (0.152) | (0.220) | (0.169) |
| Dec*White | 0.326 | -0.365 | -0.508* | -0.246 | -0.237 | 0.413* | -0.102 | -0.136 |
| | (0.371) | (0.225) | (0.295) | (0.239) | (0.225) | (0.230) | (0.293) | (0.247) |
| Mar*White | 0.698** | -0.608*** | -0.458 | -0.135 | -0.106 | 0.256 | -0.225 | -0.163 |
| | (0.345) | (0.211) | (0.284) | (0.222) | (0.216) | (0.220) | (0.288) | (0.244) |

(*Continued*)

**Table 2.** (Continued)

| | Food Acquisition Activity (= 1 if activity performed in last 14 days; = 0 otherwise) | | | | | | | |
|---|---|---|---|---|---|---|---|---|
| | Grocery Shop in Person | Grocery Shop Online | Order Meal Kit | Indoor Restaurant Dining | Outdoor Restaurant Dining | Restaurant Take-out | Visit Food Bank | Visit Farmer's Market |
| AfricanAmerican[j] | -0.428 | 0.256 | 0.304 | 0.618*** | 0.125 | -0.099 | 0.212 | 0.261 |
| | (0.356) | (0.214) | (0.294) | (0.225) | (0.224) | (0.226) | (0.287) | (0.241) |
| Dec*AfricanAmerican | 0.005 | -0.560* | -0.243 | -0.654** | -0.397 | -0.049 | -0.309 | -0.230 |
| | (0.457) | (0.298) | (0.385) | (0.312) | (0.311) | (0.312) | (0.372) | (0.331) |
| Mar*AfricanAmerican | 0.460 | -0.315 | -0.356 | -0.771 | -0.288 | -0.032 | -0.428 | -0.286 |
| | (0.462) | (0.300) | (0.396) | (0.316) | (0.323) | (0.321) | (0.398) | (0.353) |
| Constant | 1.987*** | -1.508*** | -2.800*** | -1.296*** | -1.039*** | 0.462** | -2.290*** | -1.632*** |
| | (0.357) | (0.224) | (0.332) | (0.233) | (0.221) | (0.215) | (0.351) | (0.254) |
| *Correlation Matrix* | | | | | | | | |
| Grocery Shop Online | -0.454*** | | | | | | | |
| Order Meal Kit | -0.171*** | 0.342*** | | | | | | |
| Indoor Restaurant Dining | 0.225*** | -0.103*** | 0.060* | | | | | |
| Outdoor Restaurant Dining | 0.130*** | -0.049 | 0.147*** | 0.383*** | | | | |
| Restaurant Take-out | 0.090** | 0.089*** | 0.101*** | 0.061** | 0.195*** | | | |
| Visit Food Bank | 0.032 | 0.077* | 0.272*** | 0.082** | 0.250*** | 0.062 | | |
| Visit Farmer's Market | 0.026 | 0.071** | 0.230*** | 0.151*** | 0.269*** | 0.131*** | 0.376*** | |

Notes:

[a]Month categories relative to September 2020;

[b]Relative to Male;

[c]Income categories relative to those with income less than $50,000;

[d]Age categories relative to those 55 years or older;

[e]Region categories relative to the West region;

[f]Political affiliation categories relative to Independent/Other;

[g]Relative to individuals without a bachelor's degree;

[h]Relative to individuals who do not live in a metropolitan area;

[i]Relative to individuals who do not receive nutrition assistance (e.g., SNAP, WIC);

[j]Race categories relative to all other races.

Log-likelihood of the multivariate probit estimation was -9881.025. Standard errors in parentheses. Significance is denoted by *, **, *** for 10%, 5%, and 1% levels, respectively.

significantly more likely to engage in six of the eight acquisition activities (online grocery shopping; ordering from a meal kit service; outdoor restaurant dining; restaurant take-out; visiting a food bank; and visiting a farmer's market) than individuals who were 55 years or older, all else constant. For three of these activities (ordering from a meal kit service, outdoor restaurant dining, and restaurant take-out), the gap between these two groups persisted over time, as younger individuals were more likely to engage in these activities in December 2020 and March 2021 as well. Younger individuals were also more likely to report eating indoors in December 2020 and March 2021 compared to those 55 years or older. Participants in the 35–54 years age category behaved more similarly to the 55 years and older group in September 2020 but were significantly more likely to engage in ordering from a meal kit service, indoor restaurant dining, outdoor restaurant dining, and ordering restaurant take-out relative to those 55 years and older during the December 2020 and March 2021 survey waves.

There were also differences in food acquisition behaviors based on income, education level, and nutrition assistance recipient status. Individuals in the highest income category ($100,000 or more) were significantly more likely to shop for groceries online and order from a meal kit service relative to individuals with household incomes less than $50,000, all else constant. Further, these individuals were more likely to engage in several food acquisition activities in December 2020 than their lower-income counterparts, including shopping for groceries in person, shopping for groceries online, indoor restaurant dining, and visiting a farmer's market. For education, we found that individuals with a bachelor's degree were significantly more likely to eat at a restaurant and sit outdoors and visit a farmer's market, all else constant. In December 2020, several acquisition behaviors changed for individuals with a bachelor's degree (relative to those with no degree); these individuals were less likely to engage in shopping for groceries in person and indoor restaurant dining and more likely to engage in shopping for groceries online—with the latter behavior change persisting into March 2021. Lastly, individuals who received nutrition assistance (e.g., SNAP, WIC) were significantly more likely to shop for groceries online, order from a meal kit service, and visit a food bank than non-recipients, all else constant. There was a slight decline in the probability of visiting a food bank for nutrition assistance recipients in December 2020 ($\beta$ = -0.327, p<0.10), but this would not offset the significant main effect ($\beta$ = 1.159, p<0.01) reported in Table 2.

When looking at individual food acquisition activities, we observed the most heterogeneity in restaurant dining behavior. Figs 2 and 3 illustrate the reported (raw) proportions of indoor and outdoor restaurant dining by age category and geographic region, respectively. Fig 2 reveals that rates of indoor restaurant dining increased over time for individuals ages 18–34 (36.7% in September 2020 to 50.2% in March 2021) and 35–54 years (27.5% in September 2020 to 43.7% in March 2021). Conversely, the proportion of individuals 55 years and older engaging in indoor dining dropped in December 2020 (22.4%) before rebounding in March 2021 (36.5%). Rates of outdoor dining peaked in September 2020 for individuals 55 years and older but dropped off significantly in December 2020 and March 2021 relative to those 18–34 and 35–54 years.

Fig 3 shows that rates of indoor dining were much higher in the Northeast, South, and Midwest regions of the U.S. (relative to the West region) in September 2020. The Midwest region experienced a significant drop in the proportion of participants who reported dining indoors in December 2020 while all other regions experienced increases. By March 2021, 40% or more of participants in each geographic region reported engaging in indoor restaurant dining in the prior 14 days. Outdoor restaurant dining behavior surged in the Northeast region relative to all other regions in September 2020 but experienced declines in December 2020 and March 2021. Rates of outdoor dining among Midwest participants were much lower in December 2020 and March 2021 relative to all other regions while rates of outdoor dining for individuals in the West region were fairly consistent (ranging from 30.8% to 33.0%) across the three time periods.

## Discussion

The COVID-19 pandemic has impacted nearly every aspect of daily life, including how people acquire food. The purpose of this research was to examine how food acquisition behaviors changed over the course of the COVID-19 pandemic among U.S. households. To that end, we collected survey data from U.S. consumers at three points during the pandemic (September 2020; December 2020; March 2021); each survey wave coincided with very different points in the pandemic in terms of average cases and public outlook (pessimism/optimism) toward containment of the virus.

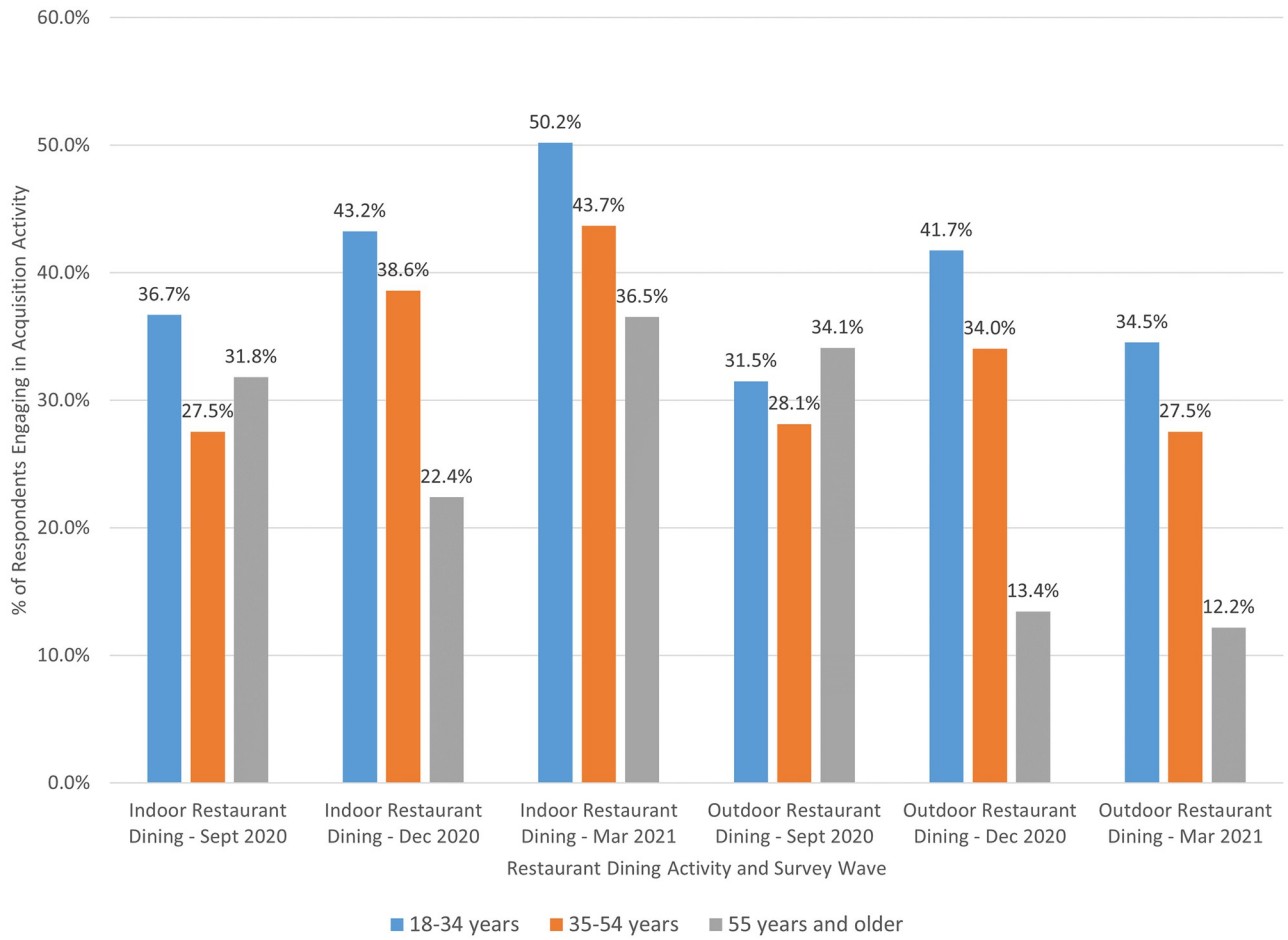

**Fig 2. Restaurant dining behavior by survey wave and age category.**

Across eight food acquisition activities, we found that shopping for groceries in person was by far the most common food acquisition activity for consumers, with 90% or more of respondents engaging in this activity in September 2020, December 2020, and March 2021. This finding suggests that brick and mortar grocery shopping is not going away in the near future, as has often been predicted, even before the COVID-19 pandemic. Interestingly, though, 30–40% of respondents also reported they shopped for groceries online in each of the three survey periods, indicating there may be a growing share of hybrid shoppers who utilize both online and in-store shopping options. This is consistent with trends reported in the grocery retailing industry since the onset of the pandemic [24, 25].

We also observed spikes in certain food acquisition activities in the December 2020 survey wave. Some of the increases observed were for what would be considered less risky activities such as shopping for groceries online and ordering from a meal kit service—an intuitive result given the surge in COVID-19 cases across the U.S. at that point in time. Other activities like visiting a food bank and visiting a farmer's market also peaked in December 2020. Some of these increases may be partially explained by seasonal effects in addition to the unique conditions created by the COVID-19 pandemic. Food banks typically experience higher demand during the holiday season, for example, and farmer's markets may have holiday-themed market days that attract more consumers. Byrne & Just [19] show there was a clear spike in Google

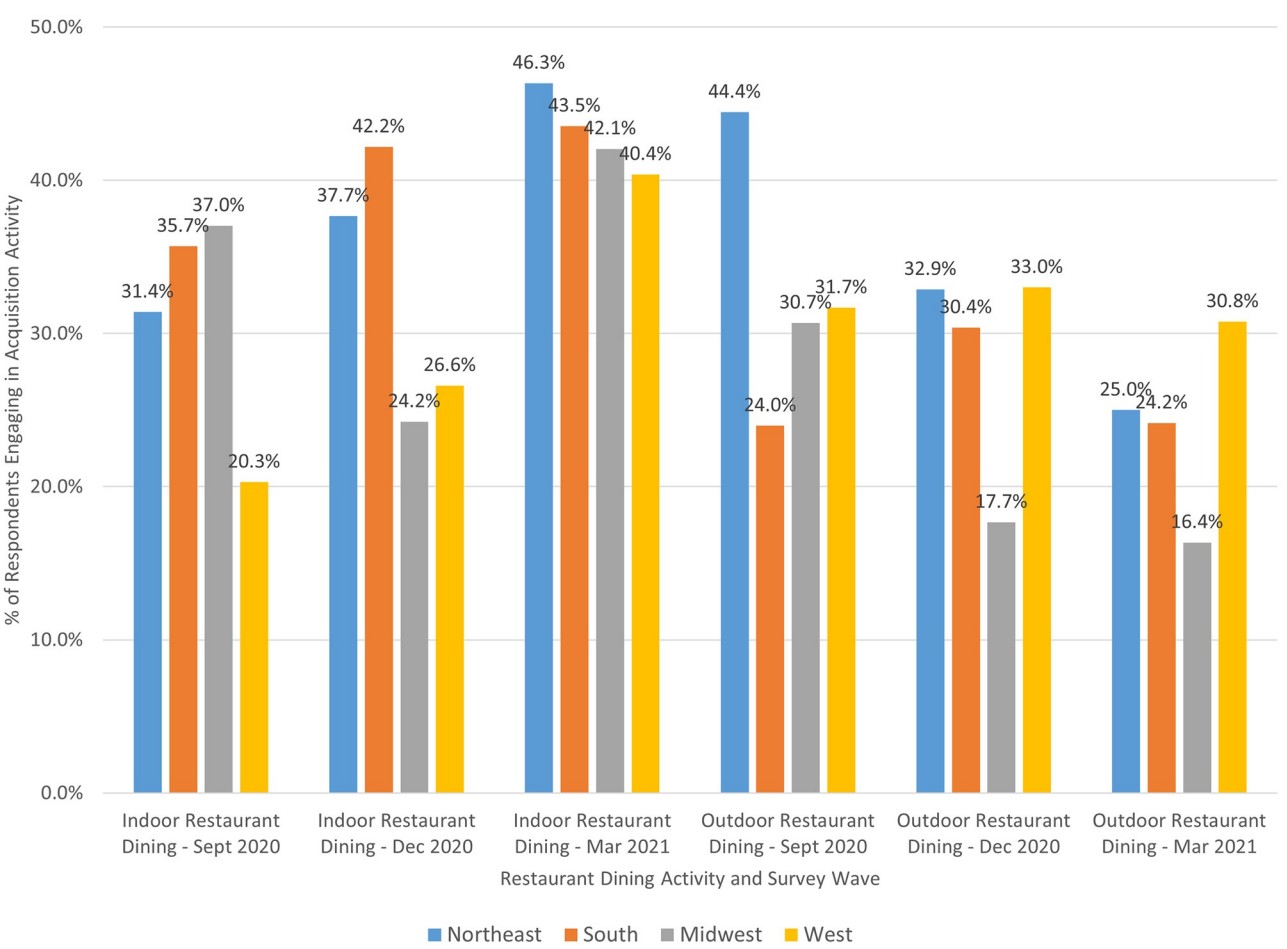

**Fig 3. Restaurant dining behavior by survey wave and geographic region.**

search data for food banks in Winter 2020, and note that unemployment rates were a better predictor of search interest compared to COVID-19 case rates. Additionally, SNAP benefits were expanded by 15% starting in January 2021 [26]. This, combined with improving unemployment rates [27], may have reduced usage of food banks in March 2021 relative to December 2020.

Restaurant dining behavior also varied over the course of the pandemic. While ordering take-out from restaurants was a fairly common activity across all time periods, indoor restaurant dining experienced a significant increase in the March 2021 survey wave as access to vaccines became more widespread. Industry data reveals that restaurant sales also experienced a big jump in March 2021—almost a 14% increase in sales from February 2021 [28]. The upward trend in sales continued over the summer months as many states eased COVID-19 restrictions, but expectations for the industry to fully recover have been tempered by the rise of the Delta and Omicron variants, as consumers are again cutting back on dining out and restaurants have reduced their hours of operation or experienced temporary closures due to the rise in COVID-19 case numbers [28].

We also find heterogeneity in food acquisition activities across households. Younger consumers (18–34 years), in particular, engaged in a wider variety of acquisition activities than their older counterparts. They were more likely to engage in online grocery shopping and

ordering from meal kit services, yet they were also more likely to engage in indoor and outdoor restaurant dining, particularly during periods that were deemed more risky (December 2020 and March 2021). These findings suggest younger consumers may have been less concerned about COVID-19 transmission or severity of illness, if contracted—which aligns with findings from the CDC [29]. Additionally, we observe that those who reported receiving nutrition assistance (e.g., SNAP, WIC) were more likely to visit a food bank—an intuitive result. Nutrition assistance recipients were also more like to engage in online grocery shopping and ordering from a meal kit service than non-recipients. At first glance, the latter results seem less intuitive; however, online shopping options were expanded during the pandemic for SNAP users, which may have increased usage [17]. In regard to meal kits, many companies offer free or heavily discounted meals for new users, so it is possible that some households utilized free trial options during the pandemic as a way to acquire more food. Newer meal kit services designed for low-income households have also entered the market in recent years, with at least one accepting SNAP benefits [30]; usage of these services likely experienced an increase during the pandemic much like the larger meal kit companies. The USDA also offered a Farmers to Family Food Box Program [31] during the pandemic that some respondents may have interpreted as a meal kit. Finally, we observe differences in restaurant dining behavior by geographic region. Some results may be partially explained by weather trends—for example, decreases in the share of consumers eating outdoors in the Northeast region in the December 2020 and March 2021 survey periods or the relatively constant rate of outdoor dining across all survey periods among consumers in the more temperate West region. Other results, though, were likely a product of COVID-19 case rates. In contrast to other regions, both indoor and outdoor restaurant dining significantly declined among Midwest consumers in the December 2020 survey period, which coincided with large spikes in COVID-19 cases in Midwestern states (relative to other regions) in November and early December 2020 [21].

This study makes important contributions to the literature, yet some limitations should be acknowledged. First, our study only assesses whether individuals engaged in food acquisition activities in a discrete (yes/no) format; we cannot speak to the frequency with which consumers engaged in these activities over the 14-day period. Second, due to the cross-sectional nature of our three surveys, we cannot track changes in behaviors for specific individuals over time but acknowledge that this would be an interesting area for future research. Lastly, our data revealed a larger proportion of straightlined responses in the December 2020 period, which resulted in more observations being excluded. This could be a function of respondents being busier during the holiday season and subsequently less attentive while taking the survey.

While our results offer new insights into household food acquisition behaviors at later points in the pandemic, there are still questions that remain. For example, as the pandemic continues to evolve and vaccinations (and boosters) become available to the youngest segments of the population, will acquisition behaviors continue to change or have consumers reached a new equilibrium in their food acquisition routines? Further, how do changes in food acquisition behaviors affect dietary quality? We leave these questions to future research.

## Conclusion

While the COVID-19 pandemic affected how many U.S. consumers acquired food, some behaviors remain unchanged. Shopping for groceries in person was near ubiquitous among consumers in all three time periods surveyed. Despite this, rates of online grocery shopping remained higher than pre-pandemic levels, indicating that online grocery shopping may be a complement, rather than a substitute, for in-person grocery shopping. For grocery retailers, it

will be important to understand how consumers utilize each mode of shopping to inform their omnichannel strategy and marketing efforts. Relative to grocery stores, our results suggest that restaurants may be more vulnerable to surges in COVID-19. Consumers consistently acquired food via restaurant take-out during each survey wave, but rates of on-site restaurant dining (indoor and outdoor) were more variable. Restaurants that heavily depend on revenue from on-site dining may want to consider how they can better diversify into take-out options (e.g., offering delivery or working with 3rd-party food delivery apps) or focus marketing efforts on younger consumers whose dining habits appear to be more resilient to changes in COVID-19 case rates.

## Supporting information

**S1 File. Data and coding file.**
(XLSX)

## Author Contributions

**Conceptualization:** Brenna Ellison, Melissa Ocepek, Maria Kalaitzandonakes.

**Data curation:** Brenna Ellison, Melissa Ocepek.

**Formal analysis:** Brenna Ellison.

**Funding acquisition:** Brenna Ellison, Melissa Ocepek.

**Investigation:** Brenna Ellison, Melissa Ocepek, Maria Kalaitzandonakes.

**Methodology:** Brenna Ellison, Melissa Ocepek, Maria Kalaitzandonakes.

**Project administration:** Brenna Ellison, Melissa Ocepek.

**Writing – original draft:** Brenna Ellison, Melissa Ocepek, Maria Kalaitzandonakes.

**Writing – review & editing:** Brenna Ellison, Melissa Ocepek, Maria Kalaitzandonakes.

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
