## [Decision Letter · Decision Letter 0]

11 May 2022

PONE-D-22-09680U.S. Household Food Acquisition Behaviors During the COVID-19 PandemicPLOS ONE

Dear Dr. Ellison,

Thank you for submitting your manuscript to PLOS ONE. After careful consideration, we feel that it has merit but does not fully meet PLOS ONE’s publication criteria as it currently stands. Therefore, we invite you to submit a revised version of the manuscript that addresses the points raised during the review process.

ACADEMIC EDITOR:

Both reviewers find the study interesting. However there are several minor comments to be addressed before the manuscript can be accepted for publication.

We look forward to receiving your revised manuscript.

Kind regards,

Hoh Boon-Peng, PhD

Academic Editor

PLOS ONE

Journal Requirements:

3. Please amend your current ethics statement to address the following concerns:

a) Did participants provide their written or verbal informed consent to participate in this study?

"This research was supported by funding from the National Institute of Food and Agriculture, U.S. Department of Agriculture, under award number ILLU-470-334 at the University of Illinois at Urbana-Champaign."

"This research was supported by funding from the National Institute of Food and Agriculture, under award number ILLU-470-334 at the University of Illinois at Urbana-Champaign (author: BE). The funders had no role in study design, data collection and analysis, decision to publish, or preparation of the manuscript."

Reviewers' comments:

Reviewer's Responses to Questions

**Comments to the Author**

1. Is the manuscript technically sound, and do the data support the conclusions?

Reviewer #1: Yes

Reviewer #2: Yes

2. Has the statistical analysis been performed appropriately and rigorously? 

Reviewer #1: Yes

Reviewer #2: N/A

3. Have the authors made all data underlying the findings in their manuscript fully available?

Reviewer #1: Yes

Reviewer #2: Yes

4. Is the manuscript presented in an intelligible fashion and written in standard English?

Reviewer #1: Yes

Reviewer #2: Yes

5. Review Comments to the Author

Reviewer #1: Authors aim to analyze eight food acquisition activities (in person grocery shopping; online grocery shopping; ordering from a meal kit service; indoor restaurant dining; outdoor restaurant dining; ordering take-out from a restaurant; visiting a food bank; and visiting a farmer’s market) at different points in the pandemic, which allows them to evaluate how food acquisition changed as case rates changed and vaccine rollouts occurred.

Abstract

OK. However, conclusions are not completely clear for me. I can see many results, but not conclusions. Authors should clearly highlight their conclusions.

Introduction.

Authors properly settled the context and pointed out their aims as well as the study’s contributions.

Methods

They collected data from three nationally representative online samples in September 2020, December 2020, and March 2021. In each cross-sectional wave of the survey, they recruited approximately 1000 U.S. consumers to complete an online survey in the Qualtrics survey platform.

They did not recruit the same individuals to participate in each wave of the survey; each sample was recruited to be representative of the U.S. population in terms of gender, age, income, and geographic region. Individuals were eligible to participate if they were 18 years of age or older and responsible for at least 50% of the grocery shopping in their household.

Survey used a very short form to analyze how households acquired foo throughout the COVID-19 pandemic. Participants were also asked to complete a series of sociodemographic questions, including questions related to gender, age, income, education, race, geographic, metropolitan status, political affiliation, and nutrition assistance recipient status.

Inclusion and exclusion criteria were defined. Finally, they evaluated changes across time and across demographics using a multivariate probit model.

Results

OK

Discussion/Conclusion

Although not mandatory, this section should be splitting into Discussion and Conclusions. This could be related with the problem I found into the Abstract. Conclusions can be drafted as a take-home message.

Reviewer #2: I would like to congratulate the authors for this study, it is very interesting. This study shows the more frequent food acquisition activities from U.S consumers during different periods of time of the COVID-19 pandemic. So, the manuscript has been written appropriately, providing literature related to the topic.

6. PLOS authors have the option to publish the peer review history of their article (what does this mean?). If published, this will include your full peer review and any attached files.

Reviewer #1: No

Reviewer #2: **Yes: **Marta Sancho Navarro

---

## [Author Response · Author response to Decision Letter 0]

8 Jun 2022

Please see attached document and cover letter.

---

## [Decision Letter · Decision Letter 1]

5 Jul 2022

U.S. Household Food Acquisition Behaviors During the COVID-19 Pandemic

PONE-D-22-09680R1

Dear Dr. Ellison,

We’re pleased to inform you that your manuscript has been judged scientifically suitable for publication and will be formally accepted for publication once it meets all outstanding technical requirements.

Kind regards,

Hoh Boon-Peng, PhD

Academic Editor

PLOS ONE

Additional Editor Comments (optional):

Reviewers' comments:

Reviewer's Responses to Questions

**Comments to the Author**

1. If the authors have adequately addressed your comments raised in a previous round of review and you feel that this manuscript is now acceptable for publication, you may indicate that here to bypass the “Comments to the Author” section, enter your conflict of interest statement in the “Confidential to Editor” section, and submit your "Accept" recommendation.

Reviewer #1: All comments have been addressed

Reviewer #2: All comments have been addressed

2. Is the manuscript technically sound, and do the data support the conclusions?

Reviewer #1: Yes

Reviewer #2: Yes

3. Has the statistical analysis been performed appropriately and rigorously? 

Reviewer #1: Yes

Reviewer #2: Yes

4. Have the authors made all data underlying the findings in their manuscript fully available?

Reviewer #1: Yes

Reviewer #2: Yes

5. Is the manuscript presented in an intelligible fashion and written in standard English?

Reviewer #1: Yes

Reviewer #2: Yes

6. Review Comments to the Author

Reviewer #1: I would like to congratulate the authors for this version of the manuscript.

Reviewer #2: I think that authors have done a good job with the revision of the manuscript. It is an interesting topic regarding to the changes in food acquisition behaviors during the COVID-19 pandemic.

7. PLOS authors have the option to publish the peer review history of their article (what does this mean?). If published, this will include your full peer review and any attached files.

Reviewer #1: No

Reviewer #2: **Yes: **Marta Sancho Navarro

---

## [Editor Report · Acceptance letter]

7 Jul 2022

PONE-D-22-09680R1 

U.S. Household Food Acquisition Behaviors During the COVID-19 Pandemic 

Dear Dr. Ellison:

I'm pleased to inform you that your manuscript has been deemed suitable for publication in PLOS ONE. Congratulations! Your manuscript is now with our production department. 

Kind regards, 

on behalf of

Dr. Hoh Boon-Peng 

Academic Editor

PLOS ONE